# Post-Mortem Extracorporeal Membrane Oxygenation Perfusion Rat Model: A Feasibility Study

**DOI:** 10.3390/ani13223532

**Published:** 2023-11-15

**Authors:** Matthias Manfred Deininger, Carl-Friedrich Benner, Lasse Johannes Strudthoff, Steffen Leonhardt, Christian Simon Bruells, Gernot Marx, Christian Bleilevens, Thomas Breuer

**Affiliations:** 1Department of Intensive and Intermediate Care, Medical Faculty, RWTH Aachen University, 52074 Aachen, Germany; gmarx@ukaachen.de (G.M.); tbreuer@ukaachen.de (T.B.); 2Department of Anesthesiology, Medical Faculty, RWTH Aachen University, 52074 Aachen, Germany; cbleilevens@ukaachen.de; 3Medical Information Technology, Helmholtz-Institute for Biomedical Engineering, RWTH Aachen University, 52074 Aachen, Germany; benner@hia.rwth-aachen.de (C.-F.B.); leonhardt@hia.rwth-aachen.de (S.L.); 4Department of Cardiovascular Engineering, Institute of Applied Medical Engineering, Medical Faculty, RWTH Aachen University, 52074 Aachen, Germany; lasse.strudthoff@rwth-aachen.de; 5Department of Anesthesia, Intensive and Emergency Medicine, Marien Kliniken, 57072 Siegen, Germany; c.bruells@mariengesellschaft.de

**Keywords:** ECMO, perfusion, post-mortem, oxygenator, rat model, Seldinger technique, surplus, 3Rs, post vivo

## Abstract

**Simple Summary:**

Animal experiments are essential for the development and optimization of innovative biomedical soft- or hardware. These technologies, not least in the field of intensive care, contribute to an enhanced quality of care and chances of survival. One prominent example that gained public attention due to the COVID-19 pandemic is extracorporeal membrane oxygenation, in short ECMO, as it supports the heart and lungs during severe disease. Even though ECMO is already in use, the components, e.g., the oxygenator that acts as an artificial lung, need further improvement. The basic functions of new devices, e.g., the oxygenation performance, do not necessarily require living animals for the evaluation; therefore, the present study aimed to develop a post-mortem rat cadaver perfusion model exclusively using surplus animals, e.g., from in-house breeding, to reduce the number of animals used for these models, according to the animal welfare principle (the 3Rs: Replace, Reduce, Refine). It was shown that the established rat cadaver model allowed organ perfusion for up to eight hours, making it a promising model for testing new biomedical technology and, thus, a welfare-friendly alternative for existing living rat perfusion models.

**Abstract:**

The development of biomedical soft- or hardware frequently includes testing in animals. However, large efforts have been made to reduce the number of animal experiments, according to the 3Rs principle. Simultaneously, a significant number of surplus animals are euthanized without scientific necessity. The primary aim of this study was to establish a post-mortem rat perfusion model using extracorporeal membrane oxygenation (ECMO) in surplus rat cadavers and generate first post vivo results concerning the oxygenation performance of a recently developed ECMO membrane oxygenator. Four rats were euthanized and connected post-mortem to a venous–arterial ECMO circulation for up to eight hours. Angiographic perfusion proofs, blood gas analyses and blood oxygenation calculations were performed. The mean preparation time for the ECMO system was 791 ± 29 s and sufficient organ perfusion could be maintained for 463 ± 26 min, proofed via angiographic imaging and a mean femoral arterial pressure of 43 ± 17 mmHg. A stable partial oxygen pressure, a 73% rise in arterial oxygen concentration and an exponentially increasing oxygen extraction ratio up to 4.75 times were shown. Considering the 3Rs, the established post-mortal ECMO perfusion rat model using surplus animals represents a promising alternative to models using live animals. Given the preserved organ perfusion, its use could be conceivable for various biomedical device testing.

## 1. Introduction

In vitro mock circulatory loops represent a well-established concept for testing, establishing and improving innovative soft- and hardware components as well as substances in biomedicine primarily in the cardiovascular context [1]. One key limitation of classic mock loop systems is that interactions with organs or whole organ systems can only be investigated to a limited extent in the artificial environment [2]. Despite recent progress in 3D printing technology, which enables the anatomical reproduction of specific structures, it fails to bridge the divide between simulated systems and actual organs, but it may serve as an additional intermediary step [3]. As a result, in vivo tests on living animals are required at an early stage of development to make further technical adjustments and optimization.

Rats are a widely used model organism and, in many cases, represent the first animal testing stage for new biomedical products [4]. However, the animal welfare principle (the 3Rs) demands researchers replace, reduce, or at least refine animal trials wherever possible. The need for animal experiments in biotechnological development, on the one hand, and the aim of a quantitative reduction in animal experiments, on the other hand, seem to be conflicting [5]. In common practice, animals used for in-house breeding services worldwide are sacrificed regularly after passing the optimal age for breeders, even though many countries have established regulations, e.g., the cascade regulation, which requires that various alternative uses of research surplus animals be evaluated before sacrificing the animal. For Germany, it was reported in 2021 that the number of animals used for experiments and surplus animals was close to a ratio of 1:1 [6].

Critical care medicine is one of the most advanced high-technology healthcare areas [7,8]. Hence, treatment is frequently based on animal experimental findings, not least for organ support [9,10]. Extracorporeal membrane oxygenation (ECMO) represents a prominent example [11], particularly during the COVID-19 pandemic, as it is a widely used clinical procedure to provide sufficient oxygenation and decarboxylation to intensive care patients with severely impaired lung function [12]. Method-wise, either two veins (veno–venous ECMO) or a vein and an artery (venous–arterial ECMO) are cannulated for ECMO implantation. In addition to pure oxygenation, the venous–arterial ECMO allows for the support or replacement of cardiac function by adding a mechanical blood pump [13].

In addition to the blood pump, a sufficient blood oxygenator is the essential component of ECMO loops. While there are international standards, like DIN EN ISO 7199 [14], for technically assessing the capabilities of oxygenators, they do not substitute the necessity for tests in living animal models. The latter are used to study the functionality of the oxygenator with simultaneous oxygen consumption and carbon dioxide production by organs, and during dynamic interactions with blood and metabolic agents [15]. Besides oxygenators, rat ECMO models are used to develop and test different parts of the ECMO circuit with a view to translational future applications in human intensive care medicine [16]. However, when used in rats up to now, ECMO systems have only been assessed in living ones [15,16].

The primary aim of this study was to establish a post-mortem perfusion model in surplus rats using a venous–arterial ECMO system, facilitating the evaluation of biomedical soft- and hardware under optimized 3Rs animal welfare conditions. The secondary objective was to investigate preliminary data on the use of the novel oxygenator (“RatOx” [15]) in this post-mortem rat ECMO model.

## 2. Materials and Methods

The experimental protocol was approved by the Institutional Animal Care and Use Committee of the RWTH Aachen University Hospital (German TSchG § 4 (3), approval code: 30280A4, approval date: 17 January 2023) and performed in accordance with German legislation governing animal studies, following the “Guide for the Care and Use of Laboratory Animals” (NIH publication, 8th edition, 2011) and the Directive 2010/63/EU on the protection of animals used for scientific purposes (Official Journal of the European Union, 2010). The experiments were carried out at the Institute for Laboratory Animal Science and Experimental Surgery of the RWTH University Hospital Aachen, Germany.

### 2.1. Animals and Housing

Four female Sprague-Dawley rats (ApoE-/-), which, as surplus breeding rats would have been sacrificed without scientific usage, were included in this study (according to the 3Rs). Rats were housed in filter-top cages with a minimum surface area of 350 cm^2^/animal, a minimum height of 18 cm and a maximum occupation density of five animals (by the husbandry requirements of annex III of EU directive 2010/63). Water, as well as a species-appropriate diet, were available ad libitum. A strict 12 h day/night rhythm was established.

### 2.2. Euthanasia

The animals were sacrificed according to the guidelines approved by the Institutional Animal Care and Use Committee of the RWTH Aachen University Hospital which is in accordance with the German Society for Laboratory Animal Science. In brief, rats were placed in a clear acryl box to induce anesthesia with a mixture of 4 vol% isoflurane (Baxter; #06497131; Unterschleißheim, Germany), until all reflexes were extinguished. Thereafter, rats were transferred into a clear box without oxygen connection, which was filled with a non-vaporized overdose of isoflurane, and were kept there until breathing stopped. The animal death was verified by clinical tests (breathlessness) first and, after a delay of five minutes, by deriving an electrocardiogram (asystole).

### 2.3. Cannulation of Vessels

For arterial cannulation, a 20-gauge peripheral intravenous catheter (PIC), and, for venous cannulation, the same but modified 20-gauge PIC (Venflon Pro Safety 20G, Becton Dickinson, Franklin Lakes, NJ, USA), were used. To ensure that the blood inflow was optimal in the venous cannula, ten holes were drilled in the catheter’s first third (Figure 1a). For better fixation on the skin by sutures, three holes were drilled in each side of the PIC wings, for both venous and arterial cannula.

Extracorporeal circulation and oxygenation were established immediately after death was confirmed, to reduce organ hypoxia and ensure sufficient organ perfusion. Thereafter, the rats were positioned supine and fixed by straps. In Step 1, the right jugular vein was cannulated using the modified PIC catheter with the Seldinger technique (Step 1, Figure 1b). After successful venous cannulation, manual perfusion, using a 20 mL syringe prefilled with 2 mL of heparinized sodium chloride (500 IE/mL), was performed (Step 2, Figure 1b), to prevent post-mortem intravascular hemostasis and coagulation in the absence of physiological blood circulation. Under continuous venous manual perfusion, cannulation of the left carotid artery was subsequently performed as the third step using the Seldinger technique (Step 3, Figure 1b).

### 2.4. ECMO Circuit

The extracorporeal circuit consisted of a roller pump (Masterflex Ismatec Reglo ICC, Avantor, Radnor, PA, USA; Figure 2I) and the corresponding pump tube (Pump Tubing, 3-Stop, PharMed BPT, 2.79 mm, Avantor, Radnor, PA, USA), which was connected via Luer Lock adapters (Luer-Lock Tubing Adapter (Male) for Flexible Tubing, PP, 3.2 mm, RCT Reichelt Chemietechnik, Heidelberg, Germany) to Heidelberg extensions. Three-way stopcocks allowed the inclusion of the oxygenator (RatOx, Institute of Applied Medical Engineering, Department of Cardiovascular Engineering, Medical Faculty, RWTH Aachen University [15]; Figure 2II). To prepare the ECMO circuit, it was flushed with 12 mL of heparinized sodium chloride (500 IU/mL), which contained an additional 10% potassium to prevent the return of spontaneous circulation. Once the rat was cannulated successfully, the ECMO circuit was connected and the roller pump was set at a speed of 20 mL per minute. To exclude cerebral perfusion through ECMO, bilateral ligation of the carotid artery (cranial to cannulation) was performed. The experiment was continued until either a hemoglobin of 1 g/dL or lower was measured, hypovolemia resulted in the suction of the venous ECMO cannula that could not be treated effectively, or the maximum experimental duration of 8 h was reached. No external heat was applied to the rat cadavers.

### 2.5. Monitoring, Laboratory and Fluid Substitution

For the entire duration of the experiment, continuous monitoring was performed, including ECG, temperature and invasive femoral blood pressure measurement. Extracorporeal venous and arterial blood gas analyses (ABL 800 flex, Radiometer, Krefeld, Germany) were performed hourly. In particular, hemoglobin was measured as an indicator for oxygen transportation capacity and for dilution factor calculation [17], lactate as an indicator for anaerobic metabolism due to circulatory impairment resulting in the reduction in oxygen delivery to organs [18], free plasma hemoglobin as a hemolysis indicator due to roller pump-induced hemoglobin damage [19], and oxygen (pO_2_) and carbon dioxide (pCO_2_) partial pressure as initial indicators of oxygenator performance [15]. Data exceeding or falling below the measuring range of the blood gas analyzer were not considered in the calculations.

For further monitoring, extracorporeal pressure measurement was applied pre- and post-oxygenator (Figure 2III). Colloidal infusion (Gelafundin ISO 4% Ecobag, B. Braun, Melsungen, Germany) as a fluid substitution was continuously administered via an infusion pump (BBraun Perfusor fm, B. Braun, Melsungen, Germany) (Figure 2IV). The running rate of the infusion pump was adjusted variably over time to avoid the aspiration of the venous cannula as a correlate of intravascular hypovolemia. Negative pressure in the venous elastic extracorporeal tubing was employed as an indicator for signs of hypovolemia. Overall, a restrictive volume administration was performed to minimize the associated dilution effects. Supporting this, a one-time application of intravenous glucocorticoids (Solu-Decortin H 250 mg, Merck Serono, Weiterstadt, Germany) was administered after successful cannulation for membrane stabilization [20].

### 2.6. Angiography

Angiographic imaging (Ziehm Vision, Ziehm Imaging, Nuremberg, Germany) was performed to visualize the perfusion achieved by ECMO. For this purpose, a 1 mL bolus of 37 °C prewarmed X-ray contrast medium (Ulravist-370, Bayer, Leverkusen, Germany) was applied extracorporeally. For visualization of the venous inflow, retrograde perfusion with X-ray contrast medium was performed with the pump stationary, as the venous angiographic phase after arterial application did not provide a sufficient image of venous perfusion.

### 2.7. Oxymetric Calculations

Gas transfer values were calculated using venous and arterial blood samples taken before and after the blood circulated through the rat cadaver. Additionally, arterial (C_a_O_2_) and venous oxygen concentration (C_v_O_2_), arteriovenous oxygen difference (avDO_2_) and oxygen extraction ratio (O_2_ER) were calculated hourly. Values for hour 8 could not be calculated as the respective oxygen saturations (sO_2_) could not be measured with the blood gas analyzer.

### 2.8. Data Collection and Analysis

The continuous data collection of ECG, temperature and all invasive pressure measurements was performed automatically using commercially available hard- and software (LabChart 8, ADInstruments, Oxford, UK). Data analysis was performed using commercially available software packages (Microsoft Excel, Microsoft 365 MSO, Version 2112, Redmond, USA; LabChart Reader 8, ADInstruments, Oxford, UK). Descriptive statistics were performed using GraphPad Prism 10 (GraphPad Software Inc., Version 10.0.2, San Diego, CA, USA). Data are shown as mean ± SD or median (range). Figure 1 and Figure 2 were created using BioRender.com. Shotcut Video Editor (Shotcut, Version 21.12.24, Meltytech, Oceanside, CA, USA) was used for the compilation of supplementary angiographic videos.

## 3. Results

### 3.1. Cannulation and Experimental Setup

Vascular cannulation for extracorporeal oxygenation and circulation was performed in three steps within 791 ± 30 s successfully in all four rats. Step 1 included the cannulation of the right jugular vein and required on average 225 ± 53 s. Thereafter, in step 2 and 3, the manual venous perfusion, venous cannula fixation and cannulation of the left carotid artery were performed, which took on average 566 ± 30 s. The ECMO circuit (Figure 2) was subsequently connected, and the rat cadavers were extracorporeally perfused for a mean of 463 ± 26 min. Two of the four rats reached the maximal experimental time of eight hours, the others reached more than seven hours each. The weight gain within the experiment was 16.1 ± 4.7% on average. The mean weight was 460 ± 183 g before and 528 ± 188 g after the experiment (the range is shown in Table 1). The primary cause of weight gain resulted from the fluid supplementation of on average 77 ± 10 mL gelafundin per animal.

Both the extracorporeal and femoral–arterial blood pressures nearly halved within the first three hours while the speed of the roller pump remained constant (femoral pressure: (0 h) 74 ± 24 mmHg; (1 h) 41 ± 11 mmHg, −45%; (2 h) 38 ± 17 mmHg, −49%; (3 h) 39 ± 13 mmHg, −47%; pre-oxygenator pressure: (0 h) 68 ± 21 mmHg; (1 h) 65 ± 24 mmHg, −4%; (2 h) 56 ± 27 mmHg, −18%; (3 h) 45 ± 11 mmHg, −44%; Appendix A). However, a mean femoral–arterial blood pressure of about 40 mmHg was maintained over the entire period in all rat cadavers for organ perfusion (mean pressure: 43 ± 17 mmHg). Organ perfusion was additionally visualized by angiographic imaging. Arterial perfusion was analyzed by native as well as digital subtraction angiography and showed sufficient distribution of contrast medium into the vessels of all major abdominal organs as well as the extremities (Figure 3a; Appendix A). Venous perfusion was visualized by retrograde contrast application and, likewise, showed the perfusion of all major abdominal organs (Figure 3b). The body temperature of the rat cadavers asymptotically approached room temperature (22 °C), which was reached in absence of an external heat supply after about 4 h (Appendix A).

### 3.2. Blood Gas Analysis Showed Blood Dilution and Hemolysis over Time

Over time, an almost linear decrease in hemoglobin (Hb) was observed (Figure 4a). The mean Hb at baseline was 8.9 ± 2.5g/dL and dropped to 1.5 ± 0.6g/dL at the end of the experiment. Given that the total amount of hemoglobin remained constant post-mortem, the decrease in Hb over time could be used as a quantification of intravascular dilution. This equals a dilution factor of six. To allow for the comparison of laboratory and oximetric parameters over time, despite high intravascular dilution, parameters were normalized to the respective Hb where appropriate. Lactate nearly doubled over the experimental time (from (0 h) 13.8 ± 2.4 mmol/L to (7 h) 22.3 ± 4.6 mmol/L) and showed, in absolute values, a gradual elevation over time. Normalized to hemoglobin, however, lactate depicted an exponential increase within the last two hours of the experiment ((0 h) 1.6 ± 0.4 mmol/L to (7 h) 16.6 ± 7.3 mmol/L per g/dL Hb). Hemolysis, measured via free plasma hemoglobin, was linearly increasing over time when normalized to total hemoglobin ((0 h) 1.2 ± 0.5% to (7 h) 25.5 ± 2.6%; (8 h) 34.5 ± 3.5% of Hb).

Baseline blood sampling was performed venously directly after the onset of extracorporeal circulation, following the post-mortem period of apneic metabolism. Hence, the baseline p_v_O_2_ showed decreased baseline p_v_CO_2_ increased values in correlation with corresponding values thereafter. Subsequently, the hourly p_v_CO_2_ (average of 24 ± 15 mmHg) and pO_2_ (p_a_O_2_: 696 ± 45 mmHg; p_v_O_2_: 113 ± 32 mmHg) values remained consistent throughout the entire duration. The p_a_CO_2_ extraction rate of the oxygenator consistently remained below the detectable range of the blood gas analyzer in all rats. Therefore, it was not possible to determine the extraction ratio for the carbon dioxide.

### 3.3. Increased Oxygen Concentration and Extraction over Time per Unit of Hemoglobin

The mean arterial oxygen concentration per gram of hemoglobin increased by 73% over time ((0 h) 1.5 ± 0.3 mL/g to (7 h) 2.6 ± 0.2 mL/g, Figure 5a) as hemoglobin concentration decreased in the same period. Oxygen concentration in venous blood also increased over time ((0 h) 0.5 ± 0.2 mL/g; (1 h) 0.9 ± 0.1 mL/g to (6 h) 1.0 ± 0.1 mL/g; (7 h) 0.9 ± 0.3 mL/g, Figure 5b), but not in as pronounced a way arterial, which is reflected in the arteriovenous oxygen difference by a marked increase between hours six to seven ((1 h) 0.75 ± 0.13 mL/g to (6 h) 1.21 ± 0.29 mL/g, *+*61%; (7 h) 1.71 ± 0.23 mL/g, +128%; Figure 5c), when Hb dropped in all rats under 3 g/dL. The increased oxygen consumption per gram of hemoglobin was also evident in the increasing oxygen extraction ratio over time ((1 h) 8 ± 2%; (6 h) 24 ± 10%, +300%; (7 h) 38 ± 7%, +475%; Figure 5d).

## 4. Discussion

In this study, we demonstrated the feasibility of establishing and performing a post-mortem ECMO circulation in rat cadavers for about eight hours. To the author’s knowledge, so far studies involving ECMO in rats have been conducted either in live rats or, at the very least, the ECMO systems were implanted while the rats were alive [16].

Following the principles of the 3Rs, this novel rat model introduced in our study represents an enhancement in terms of animal welfare. Replacement might be achieved by conducting experiments on cadavers instead of live rats. From the perspective of refinement, post-mortem experiments eliminate the risk of pain, stress or discomfort, which in experiments on living animals can never be completely ruled out, despite analgesic therapy. In terms of reduction, only rats that would otherwise be sacrificed as surplus were used, which explains the large variance in animal weight but, at the same time, illustrates that the surgical cannulation was feasible in this wide weight range without methodical adaptions. In particular, the Seldinger technique allowed reliable, rapid and atraumatic insertion of the cannulas, which was described as having an increased risk of perforation in former live rat ECMO studies [16,21].

A critical prerequisite for deriving benefits from the newly described post-mortem extracorporeal circulation method is the adequate perfusion of organs. This opens a wide range of biomedical and technological research areas, where the post-mortem rat ECMO perfusion model might be an alternative to existing trials; for example, the investigation of new pharmacological substances [22,23] as well as dosing calculations [24] on organ systems, the regulation and control of the deranged metabolism [25] or the testing of inflammatory blood filter systems [26,27] and the establishment of new ECMO components (pumps, tubing systems, membranes) [15,28,29]. As indicators of sufficient organ perfusion, besides angiographic images and femoral–arterial mean pressure primarily, blood gas analysis values were considered. In this feasibility study, we refrained from implementing interventions to stabilize factors like vascular permeability, vascular tone, metabolism and pH, apart from the single administration of glucocorticoids. Consequently, it was anticipated that the maintenance of organ perfusion could only be sustained for a limited duration, as post-mortem processes were not halted but merely decelerated. Hence, by stabilizing these values, a prolonged experimental duration might be possible. The possibility of stabilization was demonstrated recently in a successful post-mortem ECMO implantation in mice with an observational period of two hours. In this study, a normalization of blood values by the application of buffer substances and hemodialysis was demonstrated [30].

Even without stabilization measures, the current feasibility study indicated a stable circulatory perfusion of the organs and sufficient oxygen supply. The data supporting this included a stable arteriovenous oxygen ratio, venous carbon dioxide extraction and only a moderate increase in lactate within the first six hours of the experiment. Six hours also corresponds to the test period specified in DIN EN ISO 7199 for oxygenators [14].

From hour six onwards and with a mean Hb of less than 3 g/dL, an increased lactate release and a conclusively increasing oxygen extraction ratio were seen. These indicate an insufficient organ oxygen supply, not least due to progressive hemodilution with a lack of oxygen carriers. A normothermic study on an ex situ porcine liver perfusion model conclusively showed that Hb under 2–3 g/dL resulted in rising lactate [31]. Besides hemodilution, hemolysis induced by the EMCO roller pump showed another limitation concerning the long-term usage of the post-mortem model. As hemolysis is known to result in impaired vasoconstriction and endothelial function, as well as the induction of tissue hypoxia [32], the use of a pump with reduced hemolysis would be beneficial. Nonetheless, as the conditions of this post-mortem rat model are in an attenuated form comparable to the environment seen, e.g., in critically ill septic patients, this scenario represents a maximum stress test for hardware that should be used perspectival under these conditions.

Live ECMO rat models, so far, were primarily established either to investigate pathophysiological consequences for ECMO or the effects of interventions as well as drugs in rats with ECMO, or to develop ECMO hardware [15,16]. The post-mortem perfusion model presented here uses ECMO primarily as a perfusion, oxygenation and decarboxylation device, but nevertheless offers the potential to supplement or replace some uses of live ECMO rat models, as the results of the novel “RatOx” oxygenator shown in this study demonstrated.

However, this study has limitations. The study was not performed in physiological normothermia, so possibly the maximum experimental duration at 37 °C could be divergent from that reported. Given that this was a feasibility study, we used the largest oxygenator (55 layers) available, as no prior data on oxygenation and decarboxylation potential existed. The data showed, however, that in follow-up studies, the number of oxygenator layers should be reduced to achieve normocapnia and oxygenation. Regardless, the study showed that sufficient oxygenation and decarboxylation were feasible, for up to eight hours post-mortem. As mentioned previously, the major limitation of the current study is the lack of stabilization of, for example, the acid–base balance; so further studies should examine the feasibility and effects of these interventions to maximize study duration under stable organ perfusion and oxygenation. Furthermore, as this was a feasibility study, no direct comparison of the model with an ECMO in living rats was performed. In the future, this comparison might be useful to better assess the relevance of the post-mortem model as a potential substitute for live ECMO experiments.

## 5. Conclusions

In this study, we demonstrated the feasibility of post-mortem ECMO perfusion in a rat model for the first time. Using surplus rat cadavers, the model optimizes animal usage, provides an improvement according to the 3Rs of animal welfare and might represent a promising alternative to existing live rat ECMO models. Due to the preserved organ perfusion for up to six hours, it might be a promising biomedical soft- or hardware development platform.

## Figures and Tables

**Figure 1 animals-13-03532-f001:**
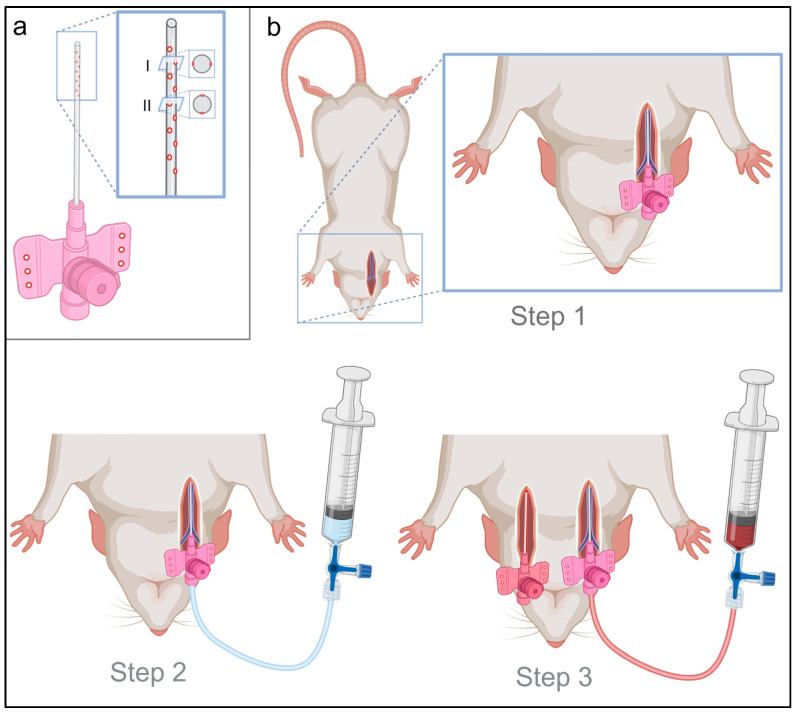
Material preparation and 3-step vessel cannulation. (**a**) The peripheral intravenous catheter (PIC) for venous cannulation was drilled with 10 holes in the first third of the PIC to ensure the best possible venous blood inflow. For this purpose, the PIC was drilled at five points in one axis bilateral (I) and five points rotated by 90° offset (II) in the other axis bilaterally. Additionally, three holes were drilled in each wing of the PIC, for optimal PIC fixation after successful cannulation. (**b**) Vessel cannulation was performed in three steps. Step 1 included blunt dissection, incision and insertion of the cannula using the Seldinger technique. Step 2 consisted of a 2 mL intravascular heparin application (500 IE/mL) and the start of manual perfusion. Step 3 included (under continuation of manual perfusion via the venous cannula) contralateral (left) dissection of the carotid artery and insertion of a cannula with an analogous surgical procedure as described in step 1.

**Figure 2 animals-13-03532-f002:**
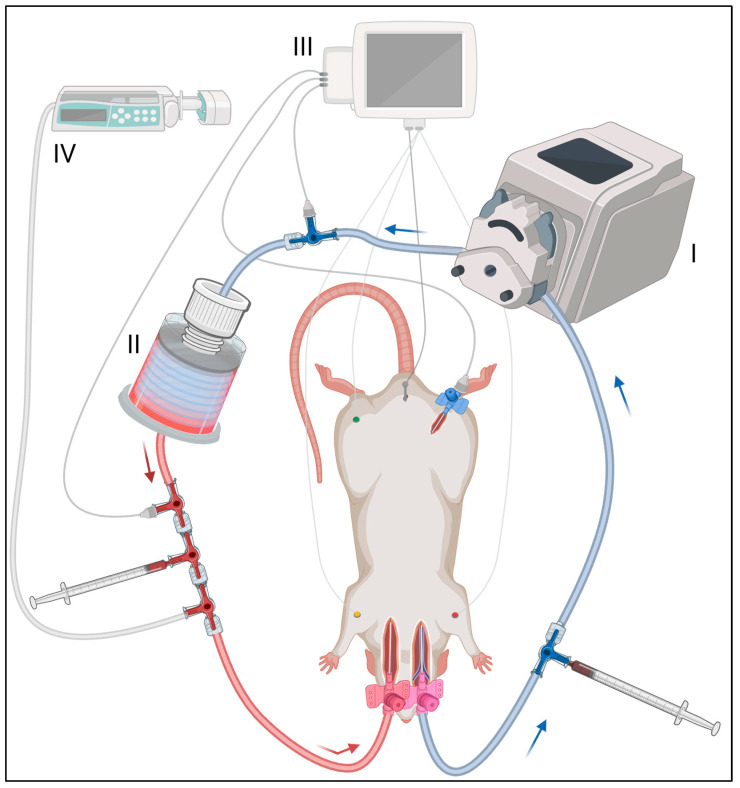
ECMO circuit setup and monitoring. The ECMO circuit included (I) a roller pump, (II) an oxygenator, (III) a measurement unit and (IV) an infusion pump. The venous cannula, as blood outflow, was connected via Heidelberg extension and three-way stopcocks with these devices. The arterial blood inflow cannula was on the left. Blood gas analysis was performed pre- and post-oxygenator (location is illustrated by 1 mL syringes). ECG, body temperature and blood pressure were measured, the latter at three locations: (1) extracorporeal pre-ECMO, (2) extracorporeal post-ECMO and (3) intra-corporeal in the right femoral artery. Gelafundin was applied via the infusion pump as a substitution for the post-mortem extravasate.

**Figure 3 animals-13-03532-f003:**
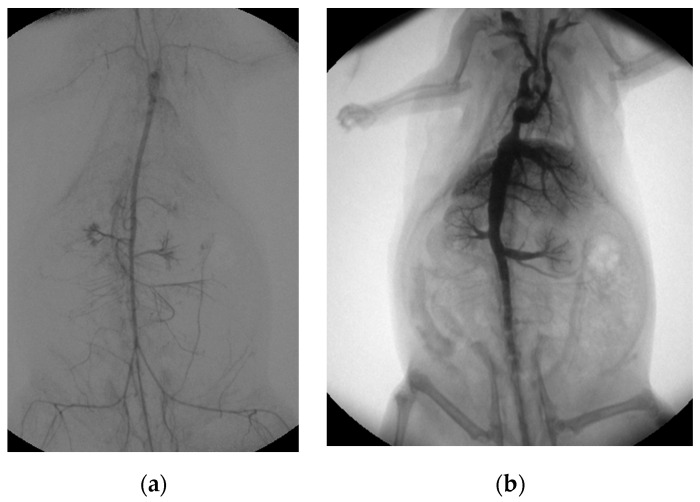
Post-mortem ECMO perfusion angiography. (**a**) shows the angiographic summation image of digital subtraction angiography after arterial contrast agent application. (**b**) illustrates the fluoroscopic image, with the venous retrograde application of contrast medium into the rat cadaver.

**Figure 4 animals-13-03532-f004:**
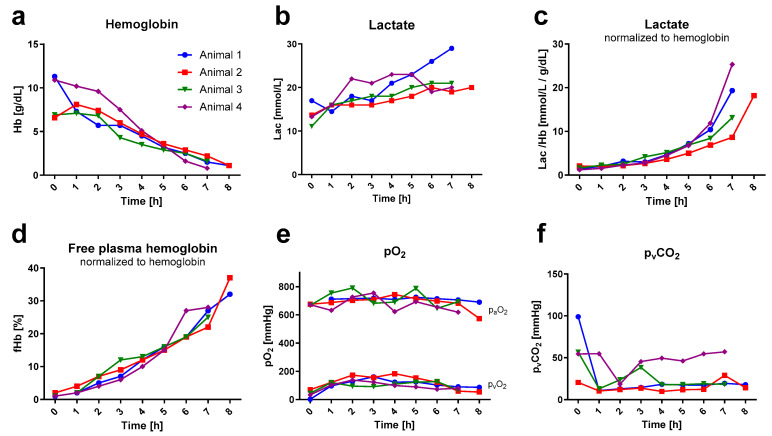
Blood gas analysis. Blood gas values (measured hourly) are illustrated over time. Data out of the blood gas analyzer’s range are not depicted. Lactate and free hemoglobin are shown in relation to hemoglobin. (**a**) depicts the hemoglobin (Hb) curve for each rat individually. (**b**) illustrates the lactate (Lac) development over time in absolute values and (**c**) relative to hemoglobin. (**d**) displays the percentage of free hemoglobin (fHb) per gram of hemoglobin. (**e**,**f**) illustrate the time course of the oxygen (arterial: p_a_O_2_; venous: p_v_O_2_) and venous carbon dioxide (p_v_CO_2_) partial pressures.

**Figure 5 animals-13-03532-f005:**
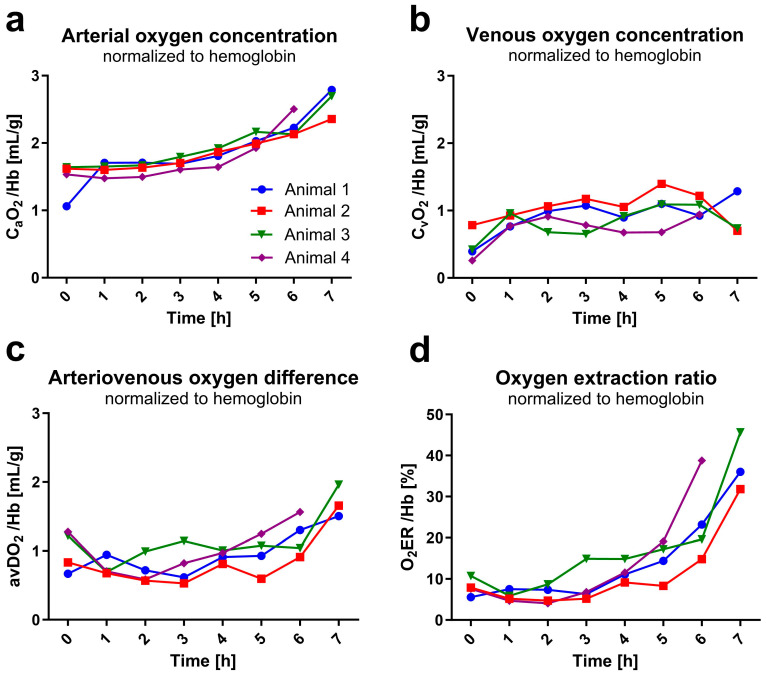
Blood and organ oxygen uptake. Values were calculated from blood gas analysis data (measured hourly) and are shown in relation to hemoglobin. Data out of blood gas analyzer’s range are not depicted. (**a**) shows arterial (C_a_O_2_) and (**b**) venous (C_v_O_2_) oxygen concentration over experimental time. (**c**) shows the arteriovenous oxygen difference (avDO_2_) resulting from the subtraction of C_a_O_2_ and C_v_O_2_. (**d**) illustrates the oxygen extraction ratio (O_2_ER) representing the ratio between oxygen uptake and delivery.

**Table 1 animals-13-03532-t001:** Demographic and surgery data.

Variable	Value
Weight (pre-experimental) [g]	374 (357–733)
Weight (post-experimental) [g]	443 (417–810)
Weight gain [%]	16.1 (10.5–21.8)
Colloid fluid infusion [mL]	76 (66–90)
Right jugular vein cannulation [s]	222 (164–292)
Left carotid artery cannulation [s]	564 (537–598)
Duration of perfusion [min]	473 (425–480)

Data are presented as median (range).

## Data Availability

The datasets used and/or analyzed during the current study are available from the corresponding author on reasonable request.

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
