# Peer review of "Post-Mortem Extracorporeal Membrane Oxygenation Perfusion Rat Model: A Feasibility Study"

_animals, 2023, doi:10.3390/ani13223532_

Round 1
Reviewer 1 Report
Comments and Suggestions for Authors
1. What is the main question addressed by the research?
The main question is if it is possible to develop an extracorporeal membrane oxygenation (ECMO) model using rat cadavers.
2. Do you consider the topic original or relevant in the field?
Does it address a specific gap in the field? What is interesting in this manuscript is that the authors propose a model by using rat cadavers instead of live animals. Another issue that is very much linked to the 3Rs, especially the Reduction of animals used for research purposes and, why not, Replacement is the use of rats which are euthanized because they are surplus breeding animals.
3. What does it add to the subject area compared with other published
material?
The manuscript could be considered as an example of different uses of
rat cadavers which are euthanized as surplus animals. This idea may
trigger the interest of researchers to use alternate methods/models in
order to evaluate equipment or systems, other than live animals.
4. What specific improvements should the authors consider regarding the
methodology? What further controls should be considered?
The main aim of the study was to develop a post-mortem ECMO perfusion
rat model. Although the setup is much more complicated than simply using
live animals, what is important is the idea of using rat cadavers. Speaking about the methodology it is interesting that the authors describe in detail the used setup which is very important for the replicability of the study
5. Are the conclusions consistent with the evidence and arguments
presented and do they address the main question posed?
I think yes
6. Are the references appropriate?
Yes, the provided reference supports the study and verifies the quality
of the obtained results
7. Please include any additional comments on the tables and figures.
Figures and tables present the obtained results as well as the setup of the procedure in a comprehensive way.
____
The aim of the manuscript was to develop a new model for post-mortem ECMO perfusion.
Among the 3Rs, we consider that the study promotes mainly Replacement rather than refinement because of the use of cadavers instead of live animals. Thus, we consider that the statement on page 9, line 317 should be rephrased accordingly.
There is a detailed description of the methodology and the monitoring protocol of the procedure. There is a detailed reference of the obtained results which support the value of the developed model. All the above are considered of great importance for either the replicability or the reproducibility of the study.
Author Response
Thank you very much for your positive review of our manuscript. Please see the attachment for a point-by-point answer.

Reviewer 2 Report
Comments and Suggestions for Authors
The authors aimed to develop a post-mortem rat perfusion model using extracorporeal membrane oxygenation (ECMO). With this model, also wanted to test the oxygenation performance of a recently developed ECMO membrane oxygnenator.
Four rats were connected post-mortem to a venous-arterial ECMO circulation for up to eight hours. Mean preparation time was about 13 min, sufficient organ perfusion could be maintained for up to eight hours, proofed via angiographic imaging and a mean femoral arterial pressure of about 43 mmHg. As only surplus animals were used, the authors suggest that this procedure could be inserted into the cascade regulation that requires that various alternative uses of research surplus animals should be evaluated before sacrificing the animal.
The aim of this study is convincing and, overall, the methods and results reported by the authors are well described. However, I have some methodological issues and concerns about the results and the discussion. I would suggest thorough revision before re-submission.
Here are more specific remarks:
· The authors nicely showed that it is possible to perform post-mortem ECMO circulation in a rat cadaver. However, what is the advantage over ECMO studies in live rats? As the number of rats would be the same – is it time, technical constrains, the detailness of the animal experiment protocol?
· The authors claim that post-mortem ECMO will serve as a viable substitute for ECMO experiments conducted on live rats. However, so far they did not directly compare outcome of ECMO with regard to organ function in living as compared to post-mortem rats. Ultimately, the quality of the organs determines the usefulness of ECMO. At least the authors should explain the similarities and differences between ECMO in alive and post-mortem rats in more detail.
· The animal`s death was verified by clinical tests (breathlessness) first and after a delay of five minutes by deriving an electrocardiogram (asystole). Nevertheless, potassium still had to be injected to prevent the return of spontaneous circulation. How was brain function monitored? Although brain function is the most sensitive, it might function on a low level while, for example, the kidneys and liver are still functioning? At least in clinical use, the patient is awake during ECMO in certain circumstances.
· Has all blood been exchanged by heparinized NaCl-solution? There must have at least some blood/hemoglobin left, because 1g/dl was used as endpoint.
· The angiographic picture with contrast only show the big body vessels but no time-series with artery first followed by venous perfusion. Was angiographic imaging sufficient to estimate preserved organ perfusion e.g., of the kidney and the brain?
· It is certainly useful to show that post-mortem ECMO is possible in different sized rats and different strains. However, specific scientific questions will certainly require rats with specific characteristics, which would counteract the use of surplus rats.
Author Response
Thank you very much for your detailled review of our manuscript. Please see the attachment for our point-by-point answer.

Reviewer 3 Report
Comments and Suggestions for Authors
This manuscript examines the feasibility of using rat cadavers to test the performance of a membrane oxygenator used to provide heart and/or lung support for critical care patients. This extracorporeal membrane oxygenation (ECMO) model was tested in four female Sprague Dawley culled as surplus breeders to reduce animal use compared to using living ECMO rat perfusion models.
The highly detailed methods and results are supported by excellent figures.
Evidence of sufficient organ perfusion was taken as angiographic imaging results, mean femoral arterial pressure of 43±17mmHg, stable arterio-venous oxygen ratio, venous carbon dioxide extraction and moderate levels of lactate produced by anaerobic metabolism. Other physiological indicators measured were oxygen (pO2) and carbon dioxide (pCO2) partial pressure for oxygenator performance, free plasma hemoglobin for hemolysis and hemoglobin for oxygen transportation capacity and dilution factor calculations.
Clearly defined experimental endpoints were provided i.e. the experiment was continued until either a hemoglobin of 1g/dl or lower was measured, hypovolemia resulted in the suction of the venous ECMO cannula that could not be treated effectively or the maximum experimental duration of 8 hours was reached.
Limitations in the model included not maintaining normal physiological values such as body temperature, acid-base balance and blood oxygenation and carbon dioxide levels. These were identified as improvements for future work.
This is a thorough study which demonstrates the feasibility of post-mortem ECMO perfusion in a rat model.
Comments on the Quality of English LanguageMinor English editing required
Line 106 ‘Rats housed in filter-top cages…’ should be ‘Rats were housed in filter-top cages…’
Line 135 ‘heparinized natrium chloride’ should be in English rather than Latin i.e. ‘heparinized sodium chloride’
Line 147 ‘Step 1 included blunt preparation…’ suggest replacing with ‘Step 1 included blunt dissection…’
Line 192 ‘Running rat of the infusion pump was adjusted variably over time…’ should be ‘The running rate of the infusion pump was adjusted variably over time…’
Line 263 This heading does not make sense ‘3.2. Laboratory results indicate for dilution and hemolysis’
Line 278 This sentence is not clear ‘Hence, the initial pvO2 showed decreased, pvCO2 increased values in correlation to steady state thereafter.’
Between lines 284 and 285 - Figure 4d Heading ‘Free plasma hemoglobine’ should be ‘Free plasma hemoglobin’
Line 319 ‘From perspective of replacement…’ should be ‘From the perspective of replacement…’
Lines 344-347 ‘Even without stabilization measures, the data of the current feasibility study showed, that stable arterio-venous oxygen ratio, venous carbon dioxide extraction and only a moderate increase in lactate was observed within the first six hours of the experiment, indicating a stable circulatory perfusion of the organs and sufficient oxygen supply.’
Suggest rewriting this as follows for clarity ‘Even without stabilization measures, the current feasibility study indicated a stable circulatory perfusion of the organs and sufficient oxygen supply. The data supporting this included a stable arterio-venous oxygen ratio, venous carbon dioxide extraction and only a moderate increase in lactate within the first six hours of the experiment.’
Author Response
Thank you very much for your positive review of our manuscript. Please see the attachment for our point-by-point answer.

Round 2
Reviewer 2 Report
Comments and Suggestions for Authors
The authors adequately addressed al concerns raised.